# Dynamic Shadow Unveils Invisible Semantics for Video Outpainting

**Ruilin Li**[1]     **Hang Yu**[1*]     **Jiayan Qiu**[2]

[1]School of Computer Engineering and Science, Shanghai University, China
[2]School of Computing and Mathematical Sciences, University of Leicester, United Kingdom
{ruilinli,yuhang}@shu.edu.cn, jq46@leicester.ac.uk

## Abstract

Conventional video outpainting methods primarily focus on maintaining coherent textures and visual consistency across frames. However, they often fail at handling dynamic scenes due to the complex motion of objects or camera movement, leading to temporal incoherence and visible flickering artifacts across frames. This is primarily because they lack instance-aware modeling to accurately separate and track individual object motions throughout the video. In this paper, we propose a novel video outpainting framework that explicitly takes shadow-object pairs into consideration to enhance the temporal and spatial consistency of instances, even when they are temporarily invisible. Specifically, we first track the shadow-object pairs across frames and predict the instances in the scene to unveil the spatial regions of invisible instances. Then, these prediction results are fed to guide the instance-aware optical flow completion to unveil the temporal motion of invisible instances. Next, these spatiotemporal guidances of instances are used to guide the video outpainting process. Finally, a video-aware discriminator is implemented to enhance alignment among dynamic shadows and the extended semantics in the scene. Comprehensive experiments underscore the superiority of our approach, outperforming existing state-of-the-art methods in widely recognized benchmarks.

## 1 Introduction

Shadows are an integral part of many natural scenes, providing rich but often ignored cues about both invisible semantics. While shadows in static images can help infer approximate categories and coarse shapes of invisible objects [44], dynamic shadows in videos offer even more, as they contain spatiotemporal information of invisible instances like unveiling the motion and actions of objects and even dynamic light sources in the scene.

Despite recent video outpainting methods [5, 8, 36] making significant progress in generating contextually matching content with coherent textures and scene structure. However, this consistency often applies to broader backgrounds rather than specific instances: objects may lack detail, exhibit distortions, or fail to preserve identity across frames, or become temporally misaligned with dynamic shadows, leading to inconsistencies with the motion and action of extended semantics.

Specifically, given an input video, conventional video outpainting methods often suffer from temporal inconsistency and action misalignment when an instance is temporarily outside the visible scene. As shown in Fig. 1(a), while the shadow moves, conventional methods tend to generate instances that remain static. In contrast, our method aligns the generated instance with the motion of the dynamic shadow, thereby preserving consistency across frames. Additionally, conventional approaches often lead to action misalignment, as shown in Fig. 1(b), where the actions of the generated instance do

---

*Corresponding author

39th Conference on Neural Information Processing Systems (NeurIPS 2025).

not correspond to the dynamic shadow in the scene. In contrast, our method captures the variations of the dynamic shadow and generates appropriate actions for the instance. Beyond addressing the aforementioned temporal inconsistency and action misalignment with dynamic shadows, our method is also capable of generating a harmonious light source, even when it is not explicitly visible in the ground truth, shown in Fig. 1(c).

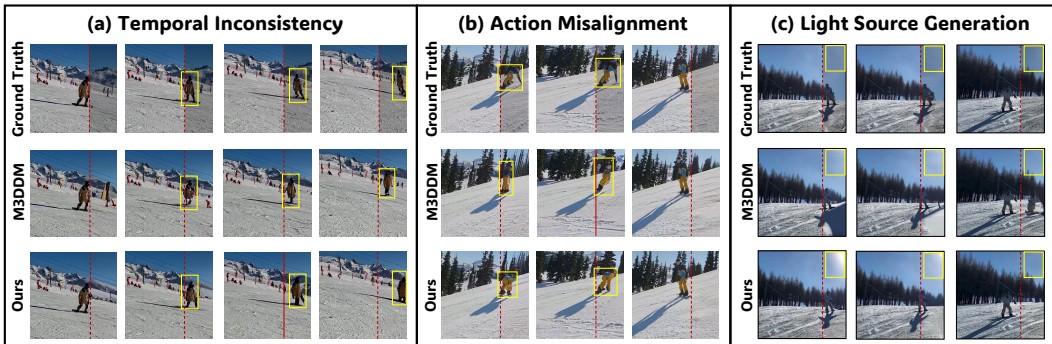

Figure 1: **Failure example of previous methods.** Many previous methods, including intensively trained models for video outpainting, often generate semantics that are temporally inconsistent with dynamic shadows and misaligned with them within the scene, as they primarily focus on maintaining coherent textures and visual consistency. In contrast, our method extends consistent semantics with dynamic shadow alignment and even generates a harmonious light across the frames.

To perform video outpainting by leveraging the hidden information in dynamic shadows, we propose a novel framework that explicitly models the associations between dynamic shadows and their corresponding instances, enabling instance-level spatiotemporal consistency even when the instances are temporarily invisible.

Specifically, we first track the shadow-instance pairs across frames and predict the instances outside the scene to unveil the spatial regions of invisible instances. Then, the prediction results are fed to guide the instance-aware optical flow completion to unveil the temporal motion of invisible instances. Next, the prediction results and the completed optical flow are used to guide the video outpainting process. Finally, a video-aware discriminator is implemented to enhance spatiotemporal alignment among dynamic shadows and the extended semantics in the scene.

Our main contributions are as follows:

- We propose a shadow-aware module that tracks and predicts the spatial region of instances, and completes optical flow in invisible areas to generate spatiotemporal guidance for outpainting.

- We introduce an instance-aware spatiotemporal guidance module that employs separate adapters for spatial predictions and completed flows to guide video outpainting, enhancing temporal consistency and semantic coherence.

- We present the first framework that explicitly leverages dynamic shadows for instance-aware video outpainting, by modeling the associations between shadows and their corresponding instances in the visible scene to unveil invisible semantics based on the cues of dynamic shadows.

## 2   Related Work

**Video Shadow Processing.** Conventional video shadow processing methods seek to track and segment dynamic shadows in complex scenes [50, 6, 2, 15]. Unlike static image shadow processing, which only focuses on individual frames, dynamic shadow processing enables the capture of spatiotemporal information in objects and light sources over time Then, a sequence of methods [11, 18, 35] is developed to enhance the tracking and segmentation of moving objects. ViShadow [38] not only detects shadows and their corresponding objects in video frames, but also continuously tracks each shadow, object, and their relationships throughout the entire video sequence. Additionally, several methods in video shadow processing, such as [38, 52], not only track shadows and objects but

also enable video editing by removing instances, all while preserving temporal coherence. However, none of the existing methods deeply look into the spatiotemporal hidden information of dynamic shadows to generate invisible semantics.

**Video Outpainting.** Conventional video outpainting methods [5, 42, 41, 30] seek to extend the content beyond its original borders based on the dynamic scene, maintaining inter-frame and intra-frame consistency in videos. Although image outpainting [4, 14, 44, 24, 19] has been extensively studied, video outpainting still needs to be fully researched. Dehan [5] proposes a modified Generative Adversarial Network (GAN) [49] for video outpainting, performing flow estimation and background estimation before integrating them into a complete result. Recently, with the powerful generative capabilities of diffusion models [29, 22, 23, 25], some diffusion-based approaches have been introduced. M3DDM [8] presents global frame-guided training with a coarse-to-fine inference pipeline to handle the artifact accumulation issue. Meanwhile, MOTIA [36] proposes a test sample-specific fine-tuning strategy to learn the patterns of each sample. However, none of these methods can take advantage of the dynamic shadows in the scene to reasonably extend the invisible areas in the video.

# 3 Method

In this section, we detail the working scheme of the proposed video outpainting framework, which comprises four sequential stages, as illustrated in Fig. 2.

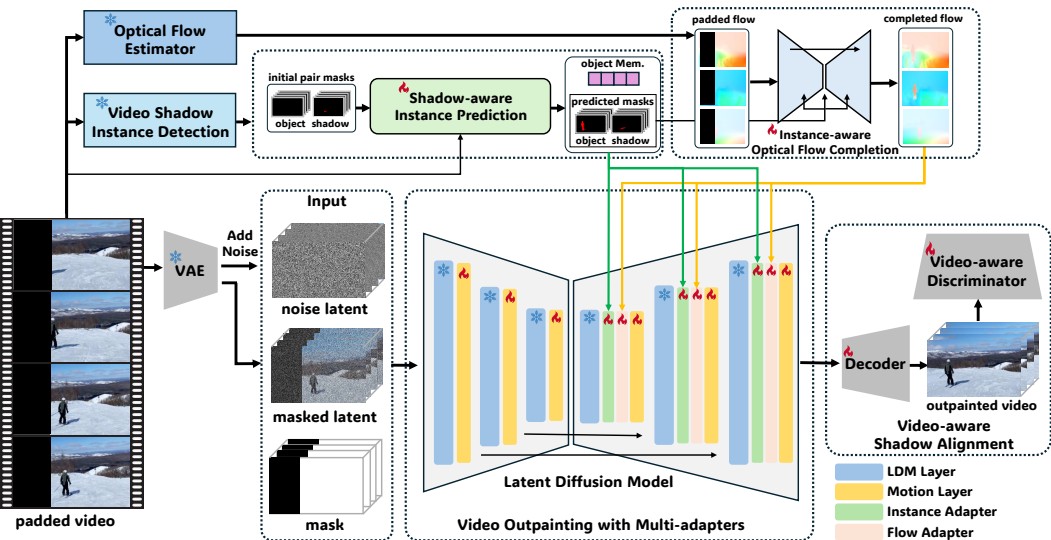

Figure 2: Illustration of the proposed framework. We first extract the shadow-object pair mask from the video and perform shadow-aware instance prediction to obtain spatial information of instances beyond the scene. This is then fed into an instance-aware optical flow completion module to generate temporal guidance for video outpainting. With the predicted spatiotemporal guidance, we perform video outpainting with two types of adapters. Finally, a video-aware discriminator is employed to enhance temporal consistency between the extended semantics and dynamic shadows.

## 3.1 Shadow-aware Instance Prediction

Tracking the mask of foreground instances in a video sequence is easy while using existing video object/instance segmentation methods [38, 27, 3]. However, predicting the masks of invisible instances that are temporarily outside the scene, either before they appear or after they leave, remains challenging due to their complex motions and actions.

To address this problem, we propose a novel shadow-aware instance prediction module that utilizes dynamic shadow information in the scene to recover the masks of instances while they are temporarily outside the visible scene.

Specifically, given the input video $V_{\text{in}} = \{I^0, I^1, \ldots, I^T\}$, where each frame $I^t \in \mathbb{R}^{H \times W \times 3}$ and $T$ is the total number of frames, we extend the video borders according to user-defined target aspect ratio to obtain the padded video $V = \{\hat{I}^0, \hat{I}^1, \ldots, \hat{I}^T\}$, where each padded frame $\hat{I}^t \in \mathbb{R}^{(H+p_h) \times (W+p_w) \times 3}$ results from adding $p_h$ pixels horizontally and $p_w$ pixels vertically to the original frame. And we define $M_{\text{vis}} \in \{0, 1\}^{H \times W}$ as the visible region mask of the video, where 1 indicates the visible area and 0 indicates the non-visible area, which is defined as:

$$V_{\text{in}} = V \odot M_{\text{vis}}, \quad V_{\text{invis}} = V \odot (1 - M_{\text{vis}}) \tag{1}$$

where $\odot$ denotes the Hadamard product, $V_{\text{invis}}$ denotes the region of $V$ that needs to be outpainted, representing the content beyond the original field of view.

We first adopt a pretrained video shadow instance detection model [38] to obtain the initial visible shadow-object pairs from the input video, which is defined as:

$$\left\{\left(M_{\text{o\_vis}}^{t,n}, M_{\text{s\_vis}}^{t,n}\right)\right\}_{t=1}^{T} = \text{Video\_Instance\_Detection}(V_{\text{in}}), \quad n = 1, 2, \ldots, N \tag{2}$$

where we assume that each frame contains $N$ shadow-object pairs, and each pair consists of an object mask $M_{\text{o\_vis}}^{t,n}$ and its corresponding shadow mask $M_{\text{s\_vis}}^{t,n}$, indicating the visible region of the object and its shadow at time $t$, respectively.

After obtaining the initial masks of visible shadow-object pairs in the scene, we aim to predict the shadow-object masks for instances that are temporarily outside the visible scene while simultaneously refining the initial tracking results, which may be inaccurate due to shadow overlap and discontinuity in complex dynamic scenes, as discussed in [44]. For simplicity, we denote the initial masks of a single shadow-object pair as $(M_{\text{o\_vis}}^t, M_{\text{s\_vis}}^t)$ at each time step.

To predict the instance masks outside the visible scene, we treat each shadow-object pair as a token in a spatiotemporal transformer [33], which is widely used in video understanding [12, 45, 51] for capturing long-range spatiotemporal coherence.

Specifically, we first apply a shared image encoder to extract high-level semantic features $F_i^t$ of each padded frame $\hat{I}^t$. And the visible object mask and the shadow mask are fed it into a mask encoder to obtain their feature embeddings $F_o^t$ and $F_s^t$, respectively, which are defined as:

$$F_i^t = \text{Image\_Encoder}(\hat{I}^t) \tag{3}$$

$$F_o^t = \text{Mask\_Encoder}(M_{\text{o\_vis}}^t), \quad F_s^t = \text{Mask\_Encoder}(M_{\text{s\_vis}}^t) \tag{4}$$

where $\text{Image\_Encoder}(\cdot)$ and $\text{Mask\_Encoder}(\cdot)$ are convolutional networks (CNNs) designed to extract features from the input image and the respective masks. Note that a standalone image encoder helps reduce redundant computation over $V$ when predicting multiple shadow-object pairs.

Next, we feed the extracted features of a single pair into the transformer for spatiotemporal fusion across frames. Specifically, for each shadow-object pair, we construct a sequence of length $T + 1$, where the 0-th input token is a learnable vector $z^0$ used to aggregate global information of this shadow-object pair, and the remaining $T$ tokens correspond to the local instance features at each time step. For each time $t$, we concatenate $F_i^t$, $F_o^t$, and $F_s^t$ along channels and feed the result into a feature encoder, obtaining the instance feature $z_t$ of time $t$ used for inter-frame fusion, which is defined as:

$$z^t = \text{Feature\_Encoder}(\text{concat}(F_i^t, F_o^t, F_s^t)), \quad t = 1, 2, \ldots, T \tag{5}$$

$$h^0, h^1, \ldots, h^T = \text{Inter-Frame\_Fusion}(z^0, z^1, \ldots, z^T) \tag{6}$$

where $\text{Feature\_Encoder}(\cdot)$ is a convolutional network designed to encode instance features of a single frame at time $t$, and $\text{Inter-Frames\_Fusion}(\cdot)$ is a transformer-based module to fuse inter-frame information of instances. The 0-th output feature $h^0$ is the aggregated global embedding of the shadow-object pair, serving as shadow-object memory in the subsequent outpainting module, while the 1st to T-th output features $h^t$ are inter-frame fused features capturing spatiotemporal correlations.

When multiple shadow-object pairs are detected, their predictions often influence each other, making it necessary to fuse spatial information across pairs. For object mask prediction, each pixel can belong to only one object. In contrast, for shadow mask prediction, a pixel may belong to several shadow regions, since shadows from different pairs may overlap.

To achieve inter-pair feature fusion within a single frame across pairs, we perform spatial fusion on the features $h_t^n$ for all pairs at each time step $t$. The temporal fused features $h_t^i$ are then passed

through another transformer for spatial fusion across shadow-object pairs, which is defined as:

$$\hat{h}^{t,1}, \hat{h}^{t,2}, \ldots, \hat{h}^{t,N} = \text{Inter-Pair\_Fusion}(h^{t,1}, h^{t,2}, \ldots, h^{t,N}), \quad t = 1, 2, \ldots, T \quad (7)$$

where each $\hat{h}^{t,n}$ is a spatiotemporal fused feature across frames and shadow-object pairs.

The spatiotemporal fused feature is passed through a lightweight mask prediction head to predict the object mask $\hat{M}_o^{t,n}$ and the shadow mask $\hat{M}_s^{t,n}$ of each shadow-object pair at each time step:

$$\hat{M}_s^{t,n}, \hat{M}_o^{t,n} = \text{Mask\_Decoder}(\hat{h}^{t,n}), \quad t = 1, 2, \ldots, T, \quad n = 1, 2, \ldots, N \quad (8)$$

where $\text{Mask\_Decoder}(\cdot)$ are convolutional networks designed to decode from each mask logit for the object and its corresponding shadow masks at time $t$.

Finally, the predicted outputs $\hat{M}_s^{t,n}$ and $\hat{M}_o^{t,n}$ are used to compute the training loss for supervising the object and shadow mask prediction in the completed scene. Additionally, the aggregated global feature $h^0$ serves as the memory embedding of a shadow-object pair and is passed to the video outpainting module to guide the recovery of the appearance of the object. Besides, the mask logits of each pair $\hat{h}^t$ are passed to the next module for guiding optical flow completion, facilitating instance-level temporal consistency outside the visible scene.

## 3.2 Instance-aware Optical Flow Completion

Pretrained flow completion modules are commonly used in video generation networks [47, 46, 13], as they provide a simpler solution for handling invisible flow compared to directly extending complex RGB content [40]. This approach leverages the inherent simplicity of flow completion to enhance video generation efficiency by using the completed flow to propagate pixels, thereby reducing the burden of video outpainting and better preserving temporal coherence.

To address this problem, we propose a novel instance-aware optical flow completion module, which recovers the instance-level optical flow outside the visible scene and provides temporal consistency guidance for the video outpainting process.

Specifically, we first use the pretrained optical flow estimation model RAFT [31] and the visible region mask $M_{\text{vis}}$ to obtain the optical flow of padded video $V$ and initialize these flows with Laplacian filling for the initialized optical flow $O_{\text{in}}$, which is defined as :

$$O_{\text{in}} = \text{Laplacian\_Filling}(\text{RAFT}(V) \odot M_{\text{vis}}) \quad (9)$$

Then, the initialized optical flow is fed into a pseudo-3D U-Net [26] encoder-decoder structure with skip connections from the encoder to the corresponding layers in the decoder, which takes the initialized optical flow $O_{\text{in}}$ and invisible region mask $(1 - M_{\text{vis}})$ and output the completed optical flow $O_{\text{out}}$, which is defined as:

$$F_{flow} = \text{Flow\_Encoder}(O_{\text{in}}), \quad F_{mask} = \text{Mask\_Encoder}((1 - M_{\text{vis}})) \quad (10)$$

$$O_{\text{out}} = \text{U-Net}(concat(F_{flow}, F_{mask}), \hat{h}^t) \quad (11)$$

where $\text{Flow\_Encoder}(\cdot)$ is a convolutional network that extracts features from the initialized optical flow and the invisible region mask. The U-Net architecture is conditioned on the spatiotemporal features $\hat{h}^t$, which enables the network to incorporate instance-level mask features, enhancing the prediction of the completed optical flow outside the scene.

Specifically, to enable optical flow completion of the padded areas with instance-awareness, we introduce a cross-attention mechanism [33] after the temporal 2D convolutions in U-Net. This mechanism allows the network to effectively incorporate conditions from the dynamic foreground, enhancing its ability to complete the optical flow, which is defined as :

$$Q = W_Q(x), \quad K = W_K(concat(\hat{h}^1, \hat{h}^2, \ldots, \hat{h}^T)) \quad (12)$$

$$V = W_V(concat(\hat{h}^1, \hat{h}^2, \ldots, \hat{h}^T)) \quad (13)$$

$$\text{Attention}(Q, K, V) = \text{softmax}\left(\frac{QK^\top}{\sqrt{d_k}}\right)V \quad (14)$$

where $x$ denotes the latent feature from the U-Net, and $\hat{h}^t$ represents the conditional mask features of time $t$. $W_Q$, $W_K$, and $W_V$ are learned linear projection layers that map inputs into the query, key, and value spaces of dimension $d_k$.

Finally, we utilize the instance logits generated by the previous module to achieve instance-aware optical flow completion, particularly in regions with sharp motion boundaries in previously invisible areas. In regions around the padded boundaries, the module promotes smoothness where there is no moving instance, while preserving sharp motion boundaries when moving instances are present.

## 3.3 Video Outpainting with Multi-adapters

To enhance the temporal coherence and spatial consistency at the instance level in video outpainting, we incorporate spatiotemporal instance conditioning and flow conditioning into a video diffusion model via a two-branch architecture, complemented by a multi-scale adapter proposed by [43], which provides different types of guidance across the U-Net blocks of latent diffusion model (LDM).

For the spatial consistency of video outpainting, we first combine the mask logit $\hat{h}_t^n$ of the t-th frame and the memory embedding $h_0^n$ for each pair $n$ at time $t$. Then, the combined features of all objects are concatenated and projected to obtain the condition for the k-th block and t-th frame in up-blocks:

$$f^{t,n} = \hat{h}^{t,n} + h^{0,n}, \quad n = 1, 2, \ldots, N \tag{15}$$

$$C_{k,t}^{mask} = \text{Mask\_Projection}(concat(f^{t,1}, f^{t,2}, \ldots, f^{t,N})) \tag{16}$$

$$x_{k,t}' = x_{k,t} + \gamma_1 \times \text{Cross\_Attn}(x_{k,t}, C_{k,t}^{mask}), \quad t = 1, 2, \ldots, T \tag{17}$$

where the parameter $\gamma_1$ controls the influence of the instance condition on the generated video And $\text{Cross\_Attn}(\cdot)$ denotes the cross-attention mechanism applied at the t-th frame and the k-th block, where $x_{k,t}$ is used to project the query, and $C_{k,t}^{mask}$ is used to project the key and value.

For the temporal conherence of video outpainting, We inject them into the U-Net with the multi-scale optical flow features extracted from $O_{out}$, We combine the latent features $x_k$ of the k-th block of the U-Net with the condition from flow features $C_k$ through element-wise addition. The combined feature is then processed by a linear layer, and its output is fed directly into the cross attention layer before each motion module in the up-blocks:

$$C_k^{flow} = \text{Flow\_Projection}(O_{out}) \tag{18}$$

$$x_k' = x_k + \gamma_2 \times \text{Cross\_Attn}(x_k, C_k^{flow}) \tag{19}$$

where the parameter $\gamma_2$ controls the influence of the flow condition on the generated video, and $x_k$ is used to project the query, and $C_k^{flow}$ is used to project the key and value.

Finally, after the diffusion process, the denoised latents are obtained by integrating the spatiotemporal conditioning signals through the U-Net. The instance and flow conditioning help guide the denoising process, ensuring both spatial consistency and temporal coherence in the generated video.

## 3.4 Video-aware Shadow Alignment Discriminator

After obtaining the denoised video latent, the latents are fed to the decoder to transform them from the latent space back to pixel space, producing the outpainted video.

Our outpainting module builds on a pre-trained image LDM [29], whose autoencoder, trained on individual images, causes flickering when applied to temporally coherent sequences, leading to shadow-object inconsistency in videos.

To address this problem, we incorporate dedicated temporal layers into the decoder, proposed in [28], to capture spatio-temporal dependencies across video frames. These layers are subsequently fine-tuned on video data, allowing the decoder to reconstruct temporally coherent textures and structures of instances within the outpainted regions.

we propose a video-aware shadow alignment discriminator that focuses on the dynamic shadow-object pairs in the scene. After obtaining the output video $V_{out}$ of decoder, extract masks of shadow-object pairs $M_o^n$ and $M_s^n$. For each shadow-object pair, we take the object mask and the shadow mask to extract mask features. For each frame, we also extract image features and then concatenate them

along the channel dimension. The resulting feature dimension is $[T, w, h, c_s + c_o + c_i]$, where $T$ is the number of frames, $w$ and $h$ are the width and height of the image feature, $c_s$ is the number of channels in the shadow mask feature, $c_o$ is the number of channels in the object mask feature, and $c_i$ is the number of channels in the image feature. Then, the result is fed into a local shadow-instance alignment video discriminator to ensure that each pair of extended semantics is visually coherent, which is defined as :

$$
\min_D V(D) = \frac{1}{2}\mathbb{E}_{x \sim p_{data}(x)}[(D(x) - b)^2] + \frac{1}{2}\mathbb{E}_{z \sim p_z(z)}[(D(V_{out}, M_o^n, M_s^n)^2],
$$
$$
\min_G V(G) = \frac{1}{2}\mathbb{E}_{z \sim p_z(z)}[(D(V_{out}, M_o^n, M_s^n) - c)^2]
$$
(20)

where $D$ denotes the alignment discriminator. $a$ and $b$ denote the ground truth real and fake labels, respectively. $c$ denotes the value that $G$ wants $D$ to believe for the fake data.

Moreover, to ensure photorealistic reconstructions, a patch-wise temporal discriminator [10] built with 3D convolutions is implemented. Additionally, the image-level shadow-instance alignment discriminators proposed by [44] are also adapted to enhance alignment between the extended semantics and their shadows. By integrating these discriminators during finetuning, the decoder learns to generate outpainted content that not only matches the spatial quality of individual frames but also preserves temporal coherence, reducing flickering, shadow misalignment, and even light source inconsistencies in the final video output.

## 4 Implementation Details

**Video Shadow Instance Detector** We directly use ViShadow [38] as our video shadow instance detector, which provides robust paired tracking of objects and their corresponding shadows, even in cases where one of them is temporarily invisible or occluded. The initial tracking results provide object–shadow relationships that serve as a heuristic to guide the tracking and prediction mask of instances outside the scene.

**Diffusion Module** Our method is built upon Stable Diffusion v1-5. The temporal modules are initialized with the weights from pretrained motion modules [9] to obtain additional motion priors. Each spatial 2D convolutional layer is followed by a temporal 1D convolutional layer, which is pretrained in text-to-video tasks on the WebVid dataset [1].

**Multi-instance Pair Prediction** Since the video shadow instance detector is limited to identifying shadow-object pairs with visible shadows, we employ YOLO [34] and SAM2 [27] to track and segment objects without visible shadows, particularly those near boundary regions. To maintain computational efficiency, we limit the number of instance pairs to a maximum of 10.

**Outpainting Strategy** For long video outpainting, M3DDM [8] adopts a hybrid coarse-to-fine pipeline, performing coarse outpainting over larger time intervals followed by fine outpainting over shorter intervals. In contrast, MOTIA [36] sequentially splits the long video into short clips with temporal overlap for outpainting. Our approach, however, takes the lifespan of instances into account. Specifically, we dynamically split the long video based on the predicted instance masks to ensure spatial consistency of instances across frames. In scenarios with multiple instances, we prioritize splitting and outpainting at segments with the highest instance overlap.

**Training Details** Our method is implemented using PyTorch [20] and trained on eight NVIDIA RTX A6000 GPUs. During training, we employ the AdamW optimizer [17] with a fixed learning rate of $1 \times 10^{-4}$ and the warm-up learning rate step is 1k. Following the training scheme of [8], the model is trained on the WebVid dataset [1] for 6 epochs with a batch size of 16, utilizing gradient accumulation to stabilize optimization under memory constraints. During the inference process, we use the PNDMScheduler [16] to guide the denoising steps in the reverse diffusion process. Also, we use 50 inference steps and a scaled linear $\beta$ schedule that starts at 0.00085 and ends at 0.012.

## 5 Experiments

In this section, we describe our experimental setup and present quantitative and qualitative comparisons, ablation studies, and a user study to validate the effectiveness of our method.

## 5.1 Datasets

We evaluate our proposed approach on two common datasets, DAVIS [21] and YouTube-VOS [39], which are widely used benchmarks for video outpainting.

**DAVIS** [21] dataset collects 150 videos in total (60 for training, 30 for validation, and 60 for testing), with each video annotated with multiple foreground instances per frame. It was originally introduced as a benchmark for Video Object Segmentation (VOS) to segment and track object instances with precise and temporally consistent annotations across video frames.

**YouTube-VOS** [39] dataset collects 4,000 high-resolution YouTube videos, totaling over 340 minutes, providing rich diversity in object appearance, motion patterns, and scene complexity. Each video is densely annotated with high-quality pixel-level masks for multiple objects in selected frames.

Additionally, since these two datasets only provide foreground object segmentation annotations and lack instance-level shadow annotations, we introduce a dataset with shadow annotations to further validate the effectiveness of our approach.

**SOBA-VID** [38] dataset collects 292 videos with a total of 7045 frames, capturing shadow-object interactions across diverse scenarios. It emphasizes varied shadow patterns, dynamic backgrounds, occlusions, and lighting conditions. The dataset is split into 232 training videos with 5863 sparsely annotated frames and 60 testing videos with 1182 densely annotated frames, resulting in 637 shadow-object pairs in total. On average, each video contains 24.1 frames and 2.2 shadow-object pairs.

## 5.2 Comparisons to Baseline Methods

**Qualitative Comparisons.** As shown in Fig. 3, we show qualitative results comparing our method to baseline approaches. M3DDM and MOTIA fail to align extended semantics with dynamic shadows. In contrast, our method generates visual content consistent with shadows, improving temporal coherence and shadow alignment. This is due to our shadow-object prediction and flow completion modules, which provide effective spatiotemporal guidance for video outpainting. Additionally, the video-aware shadow alignment discriminator enhances spatiotemporal consistency across frames.

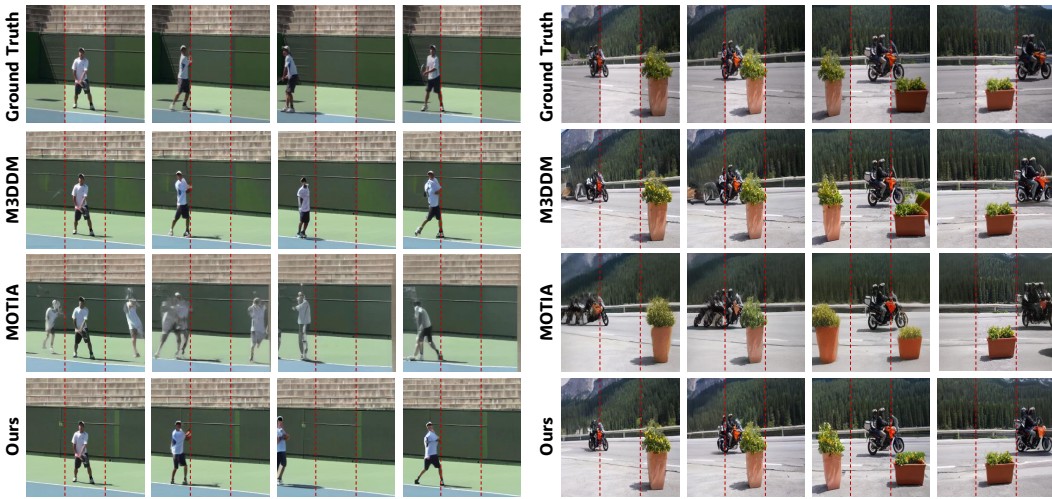

Figure 3: Qualitative comparison of video outpainting results with a mask ratio of 0.666 on the DAVIS [21] and YouTube-VOS [39] datasets.

**Quantitative Comparisons.** We compare our method with several state-of-the-art approaches on the widely used video outpainting benchmarks DAVIS [21] and YouTube-VOS [39], using four common evaluation metrics: PSNR, SSIM [37], LPIPS [48], and FVD [32]. As shown in Tab. 1, our approach achieves significant improvements over the state-of-the-art in terms of PSNR, SSIM, and FVD. However, the LPIPS score is slightly higher compared to MOTIA. Compared to our efficient zero-shot approach, MOTIA achieves better perceptual alignment and lower LPIPS scores

by fine-tuning the model on each test sample. Note that we follow the test setting of [8, 36] using mask ratios of 0.25 and 0.666 in the horizontal direction.

Table 1: Quantitative Comparisons on DAVIS [21] and YouTube-VOS [39]. ↑ means "better when higher", and ↓ indicates "better when lower". †denotes it is based on test sample-specific fine-tuning.

| Method | DAVIS [21] | | | | YouTube-VOS [39] | | | |
|---|---|---|---|---|---|---|---|---|
| | PSNR↑ | SSIM↑ | LPIPS↓ | FVD↓ | PSNR↑ | SSIM↑ | LPIPS↓ | FVD↓ |
| Dehan [5] | 17.96 | 0.6272 | 0.2331 | 363.1 | 18.25 | 0.7195 | 0.2278 | 149.7 |
| SDM [28] | 20.02 | 0.7078 | 0.2165 | 334.6 | 19.91 | 0.7277 | 0.2001 | 94.81 |
| M3DDM [8] | 20.26 | 0.7082 | 0.2026 | 300.0 | 20.20 | 0.7312 | 0.1854 | 66.62 |
| MOTIA†[36] | 20.36 | **0.7578** | **0.1595** | 286.3 | 20.25 | 0.7636 | **0.1727** | 58.99 |
| **Ours** | **20.81** | 0.7254 | 0.1842 | **234.7** | **20.32** | **0.7719** | 0.1793 | **40.78** |

## 5.3 Ablation Studies

**Ablation Study on Multi-adaption.** We further conduct a visual analysis to evaluate the effects of different adaptations on the quality of generated results, as shown in Fig. 4. It can be seen that, although all methods can generate plausible results shortly after the object becomes invisible, as the invisibility duration increases, notable differences emerge. Specifically, without the instance adapter, objects tend to exhibit artifacts or even disappear entirely. The absence of object memory embedding causes significant changes in the appearance of the object, while omitting the flow adapter leads to blurred object motion. In contrast, our method leverages dynamic shadow to generate temporally consistent objects, maintaining both appearance and motion fidelity over longer periods of invisibility.

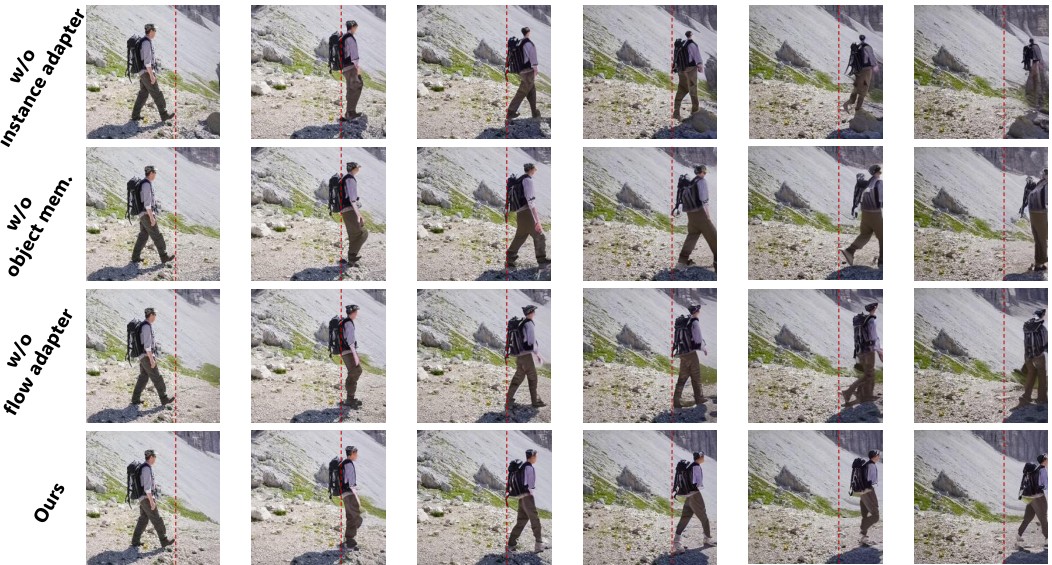

Figure 4: Visual results of multi-adaption ablation study on the DAVIS [21] dataset. The region to the right of the red line is masked out and needs to be generated.

**Ablation Study on Outpainting.** We conduct an ablation study on the YouTube-VOS [39] dataset to analyse the contribution of each module, shown in Tab. 2. The results demonstrate that both the instance and flow adapters significantly improve the video generation quality, with noticeable enhancements in PSNR, SSIM, and LPIPS. Additionally, the video-aware discriminator significantly helps reduce the FVD, further improving the overall performance of the model.

**Ablation Study on Shadow-aware Prediction.** We conduct an ablation study to assess the impact of dynamic shadow information on object mask predictions across multiple datasets, as shown in Tab. 3. Our results show that incorporating dynamic shadow improves both object and shadow mask accuracy.

Note that we introduce the SOBA-VID [38] dataset to further validate the impact of shadows, which provides annotations for both objects and shadows. In this dataset, without shadow information incorporated, objects and shadows are treated as independent instances for prediction.

Table 2: Ablation study on the YouTube-VOS [39] dataset.

| Method | PSNR↑ | SSIM↑ | LPIPS↓ | FVD↓ |
|---|---|---|---|---|
| w/ instance adapter | 19.89 | 0.7583 | 0.1881 | 64.51 |
| w/ flow adapter | 19.94 | 0.7602 | 0.1865 | 61.34 |
| w/ instance & flow adapter | 20.15 | 0.7657 | 0.1813 | 51.92 |
| w/ video-aware discriminator | 20.11 | 0.7628 | 0.1827 | 42.66 |
| Ours | **20.32** | **0.7719** | **0.1793** | **40.78** |

## 5.4 User study

We conducted a user study comparing our method against M3DDM [8] and MOTIA [36] using the YouTube-VOS [39] dataset with a horizontal mask ratio of 0.66 as source videos. We collected preferences from 40 volunteers, each evaluating 20 randomly selected result sets. The evaluation focused on two aspects: visual quality (e.g., clarity, color fidelity, and texture detail) and temporal consistency (e.g., motion smoothness, object continuity, and temporal coherence). As shown in Tab. 4, our method is preferred over both baselines in terms of both visual quality and temporal consistency.

Table 3: Ablation study of mask IoU with and without shadow incorporated across datasets.

| Method | DAVIS | YouTube-VOS | SOBA-VID |
|---|---|---|---|
| | Object | Object | Object / Shadow |
| w/o shadow | 0.75 | 0.78 | 0.65 / 0.56 |
| w shadow | 0.82 | 0.84 | 0.76 / 0.65 |

Table 4: User preference distribution (%) across three methods.

| Method | Visual Quality | Temporal Consistency |
|---|---|---|
| M3DDM [8] | 18.0% | 22.3% |
| MOTIA [36] | 25.6% | 15.7% |
| Ours | 56.4% | 62.0% |

## 6 Conclusion

In this paper, we propose a novel instance-aware video outpainting framework that leverages shadow-object associations to enhance the spatial and temporal consistency of generated content. By explicitly tracking and modeling shadow-object pairs, our method effectively unveils both the spatial region and temporal motion of instances that are invisible in the input video. The integration of spatiotemporal instance guidance with optical flow completion, along with a video-aware discriminator, ensures temporal consistency and coherent scene extension even in complex dynamic scenarios. Extensive experiments on several benchmarks demonstrate that our approach significantly outperforms existing state-of-the-art methods.

**Limitations.** Our method exploits shadow cues to improve instance generation beyond the visible scene. The informativeness of shadows depends on the angle of the light source. For example, lower angles usually cast longer shadows, which carry more semantic hints of invisible instances, such as their motion, actions, or even the location of the light source. In scenes without visible shadows, these cues are absent, and our approach falls back to standard instance-aware video outpainting.

**Societal Impacts.** Our approach can extrapolate scenes beyond the original camera view, supporting applications in film production, surveillance, and autonomous driving. However, the generated regions are synthetic and may not reflect reality, raising concerns about authenticity and trust. Following [7], we apply watermarking to label generated content and reduce the risk of misuse.

## Acknowledgements

This work is supported by National Natural Science Foundation of China (Grant No.62302287) and projects of the Shanghai Committee of Science and Technology, China (Grant No.23ZR1423500).

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
