# OpenReview forum: "Dynamic Shadow Unveils Invisible Semantics for Video Outpainting"
_NeurIPS.cc/2025/Conference — NeurIPS 2025 poster_

### Official Review · Reviewer_wJGo · 2025-06-30

**Clarity:** 3
**Significance:** 3
**Originality:** 3
**Rating:** 4
**Confidence:** 4

**Summary:**

The paper proposes the first shadow-aware video outpainting framework that uses the motion of dynamic shadows to infer and synthesize content outside the camera’s original field of view.
Shadow-aware Instance Prediction tracks object–shadow pairs across frames and predicts the spatial extent of objects that are currently outside the frame. Instance-aware Optical Flow Completion fills missing flow fields in the padded regions using cross-attention to the predicted masks, preserving object-specific motion.
A latent-diffusion outpainting network is augmented with two adapters, plus a video-aware shadow-alignment discriminator that jointly enforces spatial detail and temporal coherence.
On the DAVIS and YouTube-VOS benchmarks the method improves PSNR, SSIM and FVD over prior GAN- and diffusion-based outpainting systems, and a user study shows clear preference for visual quality and temporal consistency.

**Questions:**

How does the system resolve cases where multiple objects cast overlapping shadows?

What is the performance of training only the adapters (freezing the diffusion backbone) vs. fine-tuning the entire U-Net?

How sensitive is optical-flow completion when the camera pans quickly and shadows smear?

**Ethical Concerns:**

["NO or VERY MINOR ethics concerns only"]

**Final Justification:**

At this stage, there remains some confusion and questions about the method and performance of shadow priors. I think it is a borderline paper.

**Limitations:**

When the method comes across some special scenarios, such as camera motion, low-light, which may severely affect the visibility of the shadow, would the method still perform well? This is not validated in this manuscript.

**Paper Formatting Concerns:**

There should be punctuations after equations.

**Quality:**

3

**Strengths And Weaknesses:**

Strengths:
1. It is interesting to treat shadows as informative, temporally consistent cues, enabling the model to “unveil” otherwise invisible objects and motions.
2. extensive experiments, including quantitative metrics, qualitative comparisons, ablations, and a 40-participant user study.

Weaknesses:
1. The author should further clarify the difference between the proposed method and MOTIA[29]. MOTIA also used input-adapters in the outpainting network. It seems that the only difference is that the author introduces shadow as input prior to improving the performance of outpainting. However, observed from Tab.1, on DAVIS, the proposed method only outperforms MOTIA in PSNR and FVD (2 out of 4), and in many metrics, the two methods produce close results. The author should explain the inherent reasons.
2. The paper only presents the cases of the static camera. If the camera is moving, the relative position of the light source would change unexpectedly, leading to dramatic changes in shadow. Can the method still perform well in such cases?
3. From Tab.2, the model performs inferior to MOTIA when w/o Video-aware Shadow Alignment Discriminator. This makes the effectiveness of the shadow prior doubtful.
4. The novelty of Video-aware Shadow Alignment Discriminator is limited. As described in the manuscript, the module source from [23][10][35], making the module incremental.
5. The author should elaborate details of the user study.

---

> ### Author Rebuttal · Authors · 2025-07-31
>
> **Q1:** Difference between MOTIA and our method.
>
> **R1:** The differences between MOTIA and our method are significant.
> Specifically, MOTIA relies on test-time adaptation by training a LoRA module for each test sample, which means they need to spend considerable time training on every video during the inference stage.
> In contrast, our method does not require retraining on each video, making a comparison unfair.
> Nevertheless, the performance comparison on DAVIS remains comparable.
>
> --------------------------------------------------------------------
>
> **Q2:** The performance in scenarios with a moving camera.
>
> **R2:** As can be seen from Fig 1, 3 and 4, these several samples are from moving camera scenarios.
> And the issue of dramatic changes in shadow caused by the relative movement of the light source can be addressed by our method.
> Because our module observes the shadow-object pairs over time to infer the changing relative position of the light source.
> In addition, the spatiotemporal fusion strategy can implicitly incorporate the understanding of the changing relative position of the light source into the refinement of each shadow-object pair.
>
>
> Note that in both the DAVIS and YouTube-VOS datasets, most videos have a moving camera, and only a few are shot from a fixed position.
> To further address your concern about handling dramatic changes in shadows, we conducted an additional experiment using videos from the YouTube-VOS dataset, as can be seen from R-Tab 6.
>
> *R-Tab 6: Quantitative comparison on video in different camera motion settings from the Youtube-VOS dataset.*
> | Camera Motion | Method | PSNR ↑ | SSIM ↑ | LPIPS ↓ | FVD ↓ |
> |:---------------:|:--------:|:--------:|:--------:|:---------:|:-------:|
> |      Slow     | M3DDM  | 20.31  | 0.7351 | 0.1823  | 65.47 |
> |               | MOTIA  | 20.39  | 0.7682 | 0.1701  | 57.94 |
> |               | Ours   | 20.43  | 0.7752 | 0.1772  | 39.18 |
> |      Fast     | M3DDM  | 19.89  | 0.7197 | 0.1943  | 70.06 |
> |               | MOTIA  | 19.98  | 0.7518 | 0.1804  | 62.21 |
> |               | Ours   | 20.13  | 0.7648 | 0.1855  | 44.86 |
>
>
> The relative position of the light source and the scale of the shadow distortion are hard to quantify directly since there are no ground-truth shadow annotations in our adopted datasets.
> We therefore measure the camera motions instead, in which large camera motions between two frames are more likely with largely distorted shadows.
> The inter-frame camera motion is first measured by Structure-from-Motion (SfM).
> Then, for a single video, if its average per-frame translation is less than 5 Euclidean distance and average per-frame rotation angle is less than 5 degrees, we label it as a slow-motion video, otherwise a fast-motion one.
> As can be seen from R-Tab 2, although the fast-motion videos should be with largely distorted shadows, our method performs robustly on them.
>
> ------------------------------------------
>
> **Q3:** The effectiveness of the shadow prior in our pipeline.
>
> **R3:** Our method shows the effectiveness of the shadow prior, as can be seen from Fig 3 and Tab 1 through comparisons with M3DDM.
> As MOTIA performs test-time adaptation by training a LoRA for each test sample, which incurs large inference-time computation.
> Our method achieves better performance than M3DDM, which also does not require any test-time retraining.
> Furthermore, when our proposed components are integrated, our model outperforms MOTIA, demonstrating that the shadow prior in our method is effective.
>
> ------------------------------------------
>
> **Q4:** The novelty of video-aware shadow alignment discriminator.
>
> **R4:** It not only enhances the spatiotemporal consistency of dynamic shadow-object pairs but also enhances scene-level generation, even generating a reasonable light source, as can be seen from Fig 1(c).
> Modules like the instance adapter and flow adapter focus on capturing instance-level motion and scene-level coherent motion individually, to enhance video consistency.
> However, video generation goes beyond just handling moving instances; it also requires generating a realistic and semantic-related background, including accurate lighting and scene semantics.
> During the decoding process from latent space to video space, our discriminator contributes to generating more realistic shadows and background illumination compared to the pre-trained decoder of Stable Diffusion.
>
> ---------------------
>
> **Q5:** The details of the user study.
>
> **R5:** We conducted a user study to compare our method with M3DDM and MOTIA on the YouTube-VOS dataset with a horizontal mask ratio of 0.66.
> A total of 40 volunteers participated in the study, and each volunteer evaluated 20 randomly selected result sets.
> Participants were asked to choose the best result under each of the two criteria: visual quality (including clarity, color fidelity, and texture detail) and temporal consistency (including motion smoothness, object continuity, and temporal coherence).
>
> ------------------------------------------
>
> **Q6:** The method for resolving the overlapping shadows cast by multiple objects.
>
> **R6:** We introduce an inter-pair fusion module and a relaxed shadow mask supervision strategy to address the issue of overlapping shadows cast by multiple objects.
> Specifically, the inter-pair fusion module enables spatial information sharing across different object-shadow pairs.
> Additionally, a relaxed shadow mask supervision strategy does not penalize predictions containing connected regions of other shadows, encouraging the model to learn to complete the shadow regions of each object in complex multi-object scenarios.
>
> ---------------------
>
> **Q7:** The performance comparison of adapter-based training versus full U-Net fine-tuning in diffusion models.
>
> **R7:** Full U-Net fine-tuning outperforms adapter-based training, as can be seen from R-Tab 7.
>
> *R-Tab 7: Comparison of different fine-tuning strategies on the shrunk Stable Diffusion on Youtube-VOS.*
> | Method                 | PSNR ↑ | SSIM ↑ | LPIPS ↓ | FVD ↓ |
> |:------------------------:|:--------:|:--------:|:---------:|:-------:|
> | Only Training Adapter  | 16.74  | 0.6421 | 0.2486  | 85.32 |
> | Full U-Net Fine-tuning | 17.89  | 0.6654 | 0.2317  | 76.45 |
>
>
> However, due to computational resource constraints, we are unable to fine-tune the full U-Net of Stable Diffusion.
> Instead, we choose to freeze the backbone and train only the adapters, which still achieves competitive performance.
>
> -----------------------
>
> **Q8:** The sensitivity of optical-flow completion when the camera pans quickly and shadow smear.
>
> **R8:** Our module is not sensitive to such cases as fast camera panning and shadow smearing, as can be seen from R-Tab 8.
>
> *R-Tab 8: The sensitivity of optical-flow completion under different camera motions.*
> | Camera Motion | End-Point Error(EPE) |
> |:---------------:|:----------------------:|
> |      Slow     |          0.57        |
> |      Fast     |          0.62        |
>
> Note that, unlike video deblurring, the task of video outpainting involves clear input frames.
> Therefore, shadow smear caused by fast motion is not an issue in our task setting.
> Even under fast camera motion, our optical-flow completion still performs well due to the object-shadow mask providing instance-level spatial guidance.

---

> > ### Comment · Reviewer_wJGo · 2025-08-04
> > **Further questions**
> >
> > Thank the authors for their efforts on the rebuttal. There still remains some confusion:
> >
> > Q1:While the author states that MOTIA spent considerable time training on every video during the inference stage, they didn't provide any comparison in inference time. And more clarification in model architecture should be given, for example, "while MOTIA present xx architecture, our method is different in xxx."
> >
> > Q2: How the model discriminate whether the shadow motion is caused by object motion or camera motion?
> > Can you quantitatively define the slow and fast camera motion?
> >
> > Q3: However, the performance of w/o Video-aware Shadow Alignment Discriminator is also inferior to M3DDM. Does this mean that Video-aware Shadow Alignment Discriminator is necessary for shadow prior?
> >
> > Q6: "a relaxed shadow mask supervision strategy does not penalize predictions containing connected regions of other shadows" does not mean the model is encouraged to be aware of overlapped shadow regions of different objects.

---

> > > ### Author Response · Authors · 2025-08-06
> > >
> > > **Q1:** Comparison of inference time and more clarification in the model architecture.
> > >
> > > **A1:** Thank you for your insightful suggestion.
> > > We will provide more clarification on the architectural differences  (see A1.1) and include a detailed comparison of inference time (see A1.2) in the final version.
> > >
> > >
> > > **A1.1:** While MOTIA presents a fine-tuning strategy that learns data-specific patterns to maintain inter-frame and intra-frame consistency by retraining on each sample, our method is different in that it leverages the shadow information in the input video, which does not require retraining for each sample.
> > > Specifically, we extract and refine the shadow–object pairs in the scene and use them to guide the instance-aware outpainting process to enhance the spatiotemporal consistency of the generated video.
> > >
> > >
> > > **A1.2:** A detailed inference time comparison can be seen from R-Tab 9.
> > > MOTIA (full) refers to retraining on each sample, which consumes significant time due to data-specific adaptation.
> > > However, when MOTIA does not perform adaptation on each sample, its performance drops noticeably, as seen in MOTIA (w/o retraining).
> > >
> > >
> > > *R-Tab 9: Comparison of retraining time, the whole inference time per video, and evaluation metrics across different methods on Youtube-VOS.*
> > > |      Method         | Retrain T./Video | Inf. T./Video |  PSNR ↑   |   SSIM ↑   |  LPIPS ↓   |   FVD ↓   |
> > > | -------------------- | :--------------: | :-----------: | :-------: | :--------: | :--------: | :-------: |
> > > |  M3DDM |  N/A  |  48  s  |   20.20   |   0.7312   |   0.1854   |   66.62   |
> > > | MOTIA (full) | 427 s |  480 s  |   20.25   |   0.7636   |   0.1727   |   58.99 |
> > > | MOTIA (w/o retraining) |  N/A  |  53  s  |   19.70   |   0.7403   |   0.1846   |   70.43   |
> > > | Ours  |  N/A |55  s     |   20.32   |   0.7719   |   0.1793   |   40.78   |
> > >
> > > As mentioned in the supplementary material, we randomly selected 10 videos for outpainting at a resolution of 256×256.
> > > On a single RTX A6000 GPU, M3DDM required approximately 48 seconds per video, MOTIA took about 8 minutes, and our method completed the task in approximately 55 seconds per video.
> > >
> > >
> > > ---
> > >
> > > **Q2:** Shadow motion and quantitative definitions of slow and fast camera motion.
> > >
> > > **A2:** Our model does not explicitly discriminate whether the shadow motion is caused by object motion or camera motion. First, since object motion and camera motion often occur simultaneously, our model does not explicitly classify the cause of shadow motion. Second, by leveraging spatiotemporal feature fusion, our model learns motion patterns of instances from both spatial and temporal dimensions, capturing the differences between scenarios involving single-object motion and those involving camera motion, which leads to accurate predictions.
> > >
> > >
> > >
> > > For the quantitative definition of slow and fast camera motion, for a single video, if its average per-frame translation is less than 5 Euclidean distance and average per-frame rotation angle is less than 5 degrees, we label it as a slow-motion video, otherwise a fast-motion one.
> > > We adopt this measurement standards from Structure-from-Motion (SfM) and Novel View Synthesis (NVS) to quantify inter-frame camera motion.
> > >
> > > Since the relative position of the light source and the scale of the shadow distortion are hard to quantify directly, and there are no ground-truth shadow annotations in our adopted datasets. Specifically, large camera motions between two frames are more likely with largely distorted shadows and changing light sources.

---

> > > ### Author Response · Authors · 2025-08-06
> > >
> > > **Q3:** The effectiveness of our method without the video-aware shadow alignment discriminator.
> > >
> > > **A3:** Thank you for your insightful comment.
> > > We would like to clarify that our method is not inferior to M3DDM, when our method is without the video-aware shadow alignment discriminator, as can be seen from R-Tab 10.
> > > In fact, without this component, our method still outperforms M3DDM on key metrics such as SSIM, LPIPS, and FVD, while achieving comparable performance on PSNR with a difference of 0.05 dB.
> > >
> > > *R-Tab 10: Comparison of our method without the Video-aware Shadow Alignment Discriminator against M3DDM.*
> > > |  Method  |  PSNR ↑  | SSIM ↑ | LPIPS ↓ |  FVD ↓  |
> > > |:--:|:--:|:-:|:--:|:--:|
> > > |M3DDM  |  **20.20**   | 0.7312 | 0.1854  |  66.62  |
> > > | Ours w/o video-aware discriminator |  20.15   | **0.7657** | **0.1813**  |  **51.92**  |
> > >
> > > Our method explores dynamic shadows in the scene to improve spatiotemporal consistency. In addition to extracting shadow–object pairs to guide instance-aware outpainting, our video-aware discriminator is also necessary for shadow prior, which enhances the shadow consistency of the generated videos.
> > >
> > > --------
> > >
> > > **Q6:** The relaxed shadow mask supervision strategy.
> > >
> > > **A6:** In our early experiments, to ensure the model’s shadow completeness capability, a standard shadow supervision was applied during training. However, as the model sees more data, each pixel in the shadow region tends to be assigned to only one object, and when shadow overlaps occur, the predictions in the overlapping shadow regions become mutually exclusive for different objects. After applying the relaxed shadow mask supervision strategy, the model becomes capable of detecting overlapping shadows during training and improves the overall performance, as can be seen from R-Tab 11.
> > >
> > >
> > > *R-Tab 11: Comparison between standard shadow supervision and the relaxed shadow supervision strategy.*
> > >
> > > |Method|PSNR ↑|SSIM ↑|LPIPS ↓|FVD ↓|
> > > |:-:|:-:|:-:|:-:|:-:|
> > > | Standard Shadow Supervision | 20.22 | 0.7684 | 0.1820 | 42.94 |
> > > | Relaxed Shadow  Supervision | 20.32 | 0.7719 | 0.1793 | 40.78 |

---

> ### Author Response · Authors · 2025-08-09
>
> Dear Reviewer wJGo,
>
> We are grateful that our response has addressed your concerns and deeply appreciate your positive score for our work.
>
>
> We also thank you for your constructive feedback and will thoughtfully incorporate your comments, together with the additional analyses, into the revised manuscript.
>
> Warm regards,
>
> The Authors of Submission 19821

---

### Official Review · Reviewer_ak5T · 2025-07-01

**Clarity:** 3
**Significance:** 3
**Originality:** 3
**Rating:** 4
**Confidence:** 3

**Summary:**

This work improve the visual quality and temporal consistency by dynamic shadow information and robust object generation during occlusion. I think the motivation is reasonable but lacks further relevant verification.

**Questions:**

1. My biggest concern about this work is that it does not clearly explain the relationship between shadows and video outpainting.
2. I suspect that the most effectiveness for this work is still in the GT in the dataset, rather than the detected shadow.
3. For the visualizations in Figures 3 and 4, it is suggested that the input image should be added to make it clearer.
4. In fact, this work does not involve semantically-related content. Instances and shadow are visual features, not semantic features. Therefore, I think the hidden semantics of the above work is unconvincing.

**Ethical Concerns:**

["NO or VERY MINOR ethics concerns only"]

**Limitations:**

Although it is novel to mine temporal continuity from shadows, if the shadow part in the video is difficult to find, then this method meaning may reduce. In my opinion, generality may be the most important.

**Quality:**

3

**Strengths And Weaknesses:**

Strengths：
1. The ablation study shows that the proposed method outperforms the baselines in terms of both visual quality (e.g., clarity, color fidelity, texture detail) and temporal consistency (e.g., motion smoothness, object continuity, temporal coherence).
2. The paper demonstrates that incorporating dynamic shadow information improves the accuracy of both object and shadow mask predictions across multiple datasets.
3. The visual analysis indicates that the proposed method can generate temporally consistent objects with high fidelity in appearance and motion, even as the duration of object invisibility increases, outperforming the baselines.

Weaknesses:
Please see Questions.

---

> ### Author Rebuttal · Authors · 2025-07-31
>
> **Q1:** The relationship between shadows and video outpainting.
>
> **R1:** Even when an instance (e.g., an object) goes outside the frame, its shadow often stays in the video and with useful cues such as motion and action.
> Traditional video outpainting methods often overlook shadows in videos or treat them as noise, which is unreasonable.
> In contrast, our method explicitly models the dynamic shadows and incorporates this information as a spatiotemporal cue to enhance the spatiotemporal consistency of the generated video.
>
> ----------------------------------------------------------------
> **Q2:** The ground truth or the detected shadow.
>
> **R2:** We did not use ground truth shadow annotations, and adopted DAVIS and YouTube-VOS datasets even do not include the ground truth of shadow.
> Besides, the quality of detected shadows can impact performance, as shown in R-Tab 5.
>
>
> *R-Tab 5: Sensitivity analysis of our method to different confidence thresholds used in the shadow detector.*
>
> | Threshold | PSNR ↑ | SSIM ↑ | LPIPS ↓ | FVD ↓ |
> |:-----------:|:--------:|:--------:|:---------:|:-------:|
> |    0.2    | 19.85  | 0.7632 | 0.1865  | 42.30 |
> |    0.4    | 20.12  | 0.7685 | 0.1817  | 41.32 |
> |    0.6    | 20.32  | 0.7719 | 0.1793  | 40.78 |
> |    0.8    | 20.10  | 0.7701 | 0.1806  | 41.05 |
>
> We conducted experiments using multiple threshold settings of the video instance shadow detector.
> Our method remains robust across different thresholds, which shows our method can handle the impact of initial object-shadow detection from the video instance shadow detector.
>
> ----------------------------------------------------------------
> **Q3:** The better visualizations for the visual result.
>
> **R3:** Thank you for your valuable suggestion to improve the visualizations in Fig 3 and 4.
> We will incorporate the input images in the final version along with the current captions that explicitly indicate the masked regions to provide better visual reference and enhance clarity.
>
> ----------------------------------------------------------------
> **Q4:** The hidden semantics of dynamic shadow in the video.
>
> **R4:**  Instances such as objects and shadows are not merely visual features but also carry rich semantic information.
> Specifically, shadows are 2D projections of objects and often implicitly convey rich semantic information about objects.
> (1) In static images, shadows can unveil the category and spatial location of objects.
> (2) In videos, they further unveil the actions and motion of the objects over time.
> Therefore, modeling shadows is not just a visual consideration but also a way to capture the hidden semantics of objects in the scenes.

---

> > ### Comment · Reviewer_ak5T · 2025-08-06
> >
> > Thanks for the author's rebuttal, but the author did not fully address my concerns. I still think that the motivation of this work is soundness but naive. So I keep my opinion.

---

> > > ### Author Response · Authors · 2025-08-09
> > >
> > > **We sincerely thank the reviewer for recognizing the novelty and interest of our idea, as well as for giving us a positive score.**
> > >
> > >
> > > However, introducing dynamic shadow information into video outpainting is challenging and is not technically naive for two reasons:
> > > (a) extracting accurate shadow–object pairs is difficult due to their complex motion and varying spatial relationships in dynamic scenes;
> > > (b) generating temporally consistent video with shadow alignment is challenging due to the necessity of the instance-level spatiotemporal understanding.
> > >
> > >
> > > In addition, we conducted an experiment to compare the performance when shadows are directly introduced into baseline methods.
> > > As can be seen from R-Tab 12, simply adding shadow information to the baseline does not improve the results and even cause adverse effects.
> > >
> > > *R-Tab 12: Comparison of baselines with and without shadow information on Youtube-VOS.*
> > >
> > > | Method          | PSNR ↑ | SSIM ↑ | LPIPS ↓ | FVD ↓  |
> > > |-----------------|--------|--------|---------|--------|
> > > | M3DDM           | 20.20  | 0.7312 | 0.1854  | 66.62  |
> > > | M3DDM w/ shadow | 20.05  | 0.7260 | 0.1888  | 69.45  |
> > > | MOTIA           | 20.25  | 0.7636 | 0.1727  | 58.99  |
> > > | MOTIA w/ shadow | 20.18  | 0.7615 | 0.1742  | 59.85  |
> > > | Ours            | 20.32  | 0.7719 | 0.1793  | 40.78  |
> > >
> > >
> > > Note that for the "w/ shadow" versions, we directly adopt the detection results from the video shadow instance detector, extract the shadow and object features, concatenate them for each pair, and then feed the combined features into an adapter module, which uses cross-attention to guide the denoising process in the baseline framework.

---

> ### Author Response · Authors · 2025-08-09
>
> Dear Reviewer ak5T,
>
> We are grateful that our response has addressed your concerns and deeply appreciate your positive score for our work.
>
>
> We also thank you for your constructive feedback and will thoughtfully incorporate your comments, together with the additional analyses, into the revised manuscript.
>
> Warm regards,
>
> The Authors of Submission 19821

---

### Official Review · Reviewer_uqHS · 2025-07-02

**Clarity:** 2
**Significance:** 3
**Originality:** 3
**Rating:** 4
**Confidence:** 4

**Summary:**

This paper proposes a novel video outpainting method that enhances the temporal and spatial consistency of generated content by incorporating the prior relationship between instances and their dynamic shadows into an adapter. Conventional video outpainting methods often suffer from issues such as temporal inconsistency, motion mismatch, and a lack of instance-level modeling when processing dynamic scenes. The authors innovatively propose an auxiliary feature generation pipeline that effectively reveals the spatial regions and temporal motion of unseen instances in the input video by explicitly tracking and modeling shadow-object pairs. Experimental results demonstrate that the proposed method outperforms existing state-of-the-art methods across multiple evaluation metrics on the DAVIS and YouTube-VOS datasets, exhibiting particularly strong performance in maintaining the temporal consistency and spatial coherence of objects.

**Questions:**

1. The proposed method introduces several modules to generate auxiliary features, which are then injected into the diffusion model via an adapter. How does the computational overhead (both for training and inference) of this approach compare to the methods it is benchmarked against?
2. When multiple instances are detected, how is the global feature $h^t$ derived from the individual instance features $h^{t,n}$? For instance, are they concatenated, summed, or aggregated in some other manner?
3. Could the authors elaborate on the design of $z^0$? Is a unique latent code $z^{0,n}$ assigned to each object class? If not, how is $h^{0,n}$ derived from a single $z^0$, and how is the Inter-Pair Fusion subsequently computed across different instances?

**Ethical Concerns:**

["NO or VERY MINOR ethics concerns only"]

**Final Justification:**

The authors' partially resolved my concerns. I will raise my rating to 4.

**Limitations:**

yes

**Paper Formatting Concerns:**

The paper's formatting basically meets the requirements.

**Quality:**

4

**Strengths And Weaknesses:**

**Strengths**
1. The method explicitly incorporates prior knowledge about instances and shadows, which offers good interpretability.
2. The method is well-designed, with four modules that have distinct functions and collaborate effectively to form a complete solution. The authors validate the effectiveness of each component through comprehensive ablation and user studies.
3. Video outpainting is an important problem in the field of computer vision with broad application potential.
4. The experimental results demonstrate that the proposed method significantly outperforms existing state-of-the-art approaches across multiple metrics.
5. The paper is the first to explicitly propose the use of dynamic shadow information for video outpainting, presenting a novel research perspective.

**Weaknesses**
1. The method's applicability is constrained by the availability of shadow information. Its performance may be limited in scenarios where distinct shadows are not present.
2. The notation for superscripts and subscripts is somewhat confusing due to inconsistent use and omissions without clear explanation.

---

> ### Author Rebuttal · Authors · 2025-07-31
>
> **Q1:** The generalizability of the method to videos with or without shadows.
>
> **R1:** Our method performs well on videos both with and without shadows, as can be seen from R-Tab 3.
>
> *R-Tab 3: Quantitative comparison on videos with and without shadows from the Youtube-VOS dataset.*
> | Shadow Presence | Method | PSNR ↑ | SSIM ↑ | LPIPS ↓ | FVD ↓ |
> |:-----------------:|:--------:|:--------:|:--------:|:---------:|:-------:|
> | With Shadows    | M3DDM  | 20.19  | 0.7330 | 0.1844  | 66.54 |
> |                 | MOTIA  | 20.24  | 0.7639 | 0.1732  | 58.76 |
> |                 | Ours   | 20.35  | 0.7756 | 0.1762  | 39.62 |
> | Without Shadows | M3DDM  | 20.22  | 0.7315 | 0.1857  | 66.75 |
> |                 | MOTIA  | 20.26  | 0.7634 | 0.1725  | 59.20 |
> |                 | Ours   | 20.28  | 0.7661 | 0.1833  | 42.30 |
>
> Note that, for a single video, if the shadow area is less than 10\% of the single frame size and lasts for less than 20\% of the video duration, we label it as a video without shadows, otherwise a video with shadows.
>
> When shadows are absent (i.e., the shadow mask is all zeros), our method still performs conventional outpainting, which indicates that our model is more general, rather than specific to scenarios with shadows.
>
> -----------------------------
>
> **Q2:** The computational overhead of the proposed method relative to baselines in training and inference.
>
> **R2:** Our method offers a good trade-off between speed and quality during inference, as can be seen from R-Tab 4.
>
> *R-Tab 4: Comparison of model size, trainable parameters, and inference time across different methods on Youtube-VOS.*
> | Method | Inf. Time/Video | Params | Trainable Params | PSNR ↑ | SSIM ↑ | LPIPS ↓ | FVD ↓ |
> |:--------:|:-----------------:|:--------:|:------------------:|:--------:|:--------:|:---------:|:-------:|
> | M3DDM  |       48 s      | 1382 M |       1299 M     | 20.20  | 0.7312 | 0.1854  | 66.62 |
> | MOTIA  |       480 s     | 2254 M |       12 M       | 20.25  | 0.7636 | 0.1727  | 58.99 |
> | Ours   |       55 s      | 1593 M |       92 M       | 20.32  | 0.7719 | 0.1793  | 40.78 |
>
> Because the training code for M3DDM has not been publicly released, we have re-implemented it for testing.
> And MOTIA requires training a LoRA for each test-time sample, which incurs significant per-instance computational cost during inference preparation.
> As mentioned in the supplementary material, we randomly selected 10 videos for outpainting at a resolution of 256×256.
> On a single RTX A6000 GPU, M3DDM required approximately 48 seconds per video, MOTIA took about 8 minutes, and our method completed the task in approximately 55 seconds per video.
>
> -----------------------------
>
> **Q3:** The way to derive the global feature $h^t$ of a single frame.
>
> **R3:** The global feature $h^t$ is obtained by concatenating individual instance features $h^{t,n}$, as can be seen from Eq 16.
> In addition, the number of instance pairs is capped at a fixed maximum of 10 to ensure consistent input dimensions, as detailed in the supplementary material.
>
> -----------------------------
> **Q4:** The design of the learnable vector $z^0$ in the inter-frame fusion module.
>
>
> **R4**: This learnable vector $z^0$ is shared by each pair to extract their global embedding $h^0$ individually.
> Such an aggregation structure is similar to that used in Vision Transformer (ViT)[1], which has been demonstrated to be effective.
> This learned unique latent code $z^{0,n}$ is assigned to each shadow-object pair to derive the global embedding $h^{0,n}$.
> Then, to avoid confusion across different pairs, the global embedding $h^0$ of each pair is not involved in the inter-pair fusion process, as can be seen from Eq 7.
>
> [1] "An image is worth 16x16 words: Transformers for image recognition at scale." In the International Conference on Learning Representations (ICLR) 2021.

---

> > ### Comment · Reviewer_uqHS · 2025-08-08
> >
> > I thank the authors for their rebuttal, which partially resolved my concerns. I will raise my rating to 4.

---

> ### Author Response · Authors · 2025-08-06
>
> Dear Reviewer uqHS,
>
> We hope that our response has addressed the concerns raised in your feedback, and we sincerely appreciate your decision to update the score accordingly.
>
> If you have any additional concerns or would like to continue the discussion, we would be happy to engage further.
>
> Thank you once again for your thoughtful and constructive feedback.
>
> Best regards,
>
> The Authors of Submission 19821.

---

> ### Author Response · Authors · 2025-08-09
>
> Dear Reviewer uqHS,
>
> We are grateful that our response has addressed your concerns and deeply appreciate your decision to increase the score for our work.
>
> We also thank you for your constructive feedback and will thoughtfully incorporate your comments, together with the additional analyses, into the revised manuscript.
>
> Warm regards,
> The Authors of Submission 19821

---

### Official Review · Reviewer_G7tk · 2025-07-04

**Clarity:** 3
**Significance:** 4
**Originality:** 3
**Rating:** 5
**Confidence:** 3

**Summary:**

This paper proposes a new framework for video outpainting that makes innovative use of dynamic shadows to infer and reconstruct semantically consistent content in scenes where objects temporarily disappear from view. The authors design a multi-stage architecture that includes modules for predicting object-shadow masks, completing instance-aware optical flow, and guiding a latent diffusion model with spatio-temporal priors. The system also introduces a video-aware discriminator aimed at aligning generated content with the motion and structure implied by dynamic shadows. Experiments on widely used benchmarks show competitive improvements over several recent baselines, both in terms of visual quality and temporal coherence.

**Questions:**

(1) How well does the method generalize to real-world videos where shadows are less prominent or heavily distorted (e.g., indoor lighting, artificial light sources)? It would be helpful to see some quantitative or qualitative evaluation under such conditions.
(2) Have you considered how sensitive the overall pipeline is to the quality of the initial object-shadow detection? If this module fails or produces imprecise masks, what are the typical failure modes of the system?
(3) In your architecture, different components (e.g., instance adapter, flow adapter) are introduced in a modular way. Are these modules jointly trained end-to-end, or is there a staged training process?

**Ethical Concerns:**

["NO or VERY MINOR ethics concerns only"]

**Final Justification:**

The authors' response partially addressed my questions. There are still some minor issues not fully answered, but they do not change my rating.

**Limitations:**

While the method demonstrates impressive performance on benchmark datasets, there are notable limitations that deserve further attention. First, the reliance on high-quality dynamic shadows makes the system less generalizable to scenes where such shadows are weak, inconsistent, or altogether missing. This dependency may reduce the method’s effectiveness in many indoor environments or scenarios involving multiple light sources. Second, the approach assumes that shadow-object associations can be reliably extracted, which might not hold in crowded scenes, strong occlusion, or when shadows are cast on complex textured backgrounds. These assumptions could limit the model's robustness in unconstrained or in-the-wild video content.

**Paper Formatting Concerns:**

No major formatting concerns.

**Quality:**

3

**Strengths And Weaknesses:**

One of the most compelling aspects of this work lies in its central idea: treating dynamic shadows not as noise but as informative signals that reflect underlying instance motion. This perspective departs meaningfully from prior approaches, which tend to overlook or discard such visual phenomena. The technical formulation is coherent and thoughtfully designed, especially the integration of shadow-based instance priors into a diffusion-based generation pipeline. The empirical results are strong, and the ablation studies offer reasonable support for the importance of each module in the system.

On the other hand, the method’s reliance on accurate shadow detection raises questions about its robustness in varied lighting conditions or scenes where shadows are faint or occluded. The proposed approach also appears to require relatively heavy computational resources, which may limit its scalability to longer or higher-resolution videos. In terms of evaluation, while the reported metrics show clear improvements, the absence of statistical uncertainty estimates (e.g., variance or confidence intervals) makes it difficult to assess how consistent these gains are across different runs or datasets. Lastly, while the architectural design is detailed, it is not always easy to distinguish which parts are trained jointly and which are fixed, which could hinder reproducibility for less experienced practitioners.

---

> ### Author Rebuttal · Authors · 2025-07-31
>
> **Q1:** The generalizability of the method to real-world videos.
>
> **R1:** Our method performs well on real-world videos because shadows commonly exist in natural scenes.
> As can be seen in the supplementary material, videos from real-world datasets like DAVIS and YouTube-VOS have an average of 0.9 and 1.2 shadow-object pairs per video, with each pair lasting around 38.5 frames in DAVIS and 24.2 frames in YouTube-VOS.
> Moreover, for 76\% of videos with shadow-object pairs, the object becomes invisible while its shadow is still visible.
>
>
> It can be seen from R-Tab 1 and R-Tab 2 that our method performs robustly on the cases with less prominent shadows (R-Tab 1) and heavily distorted shadows (R-Tab 2). This is because our proposed spatio-temporal fusion strategy alleviates the effect of low-quality shadow detection in single frames by considering the entire video.
>
>
> *R-Tab 1: Sensitivity analysis of our method to different confidence thresholds used in the shadow detector.*
>
> | Threshold | PSNR ↑ | SSIM ↑ | LPIPS ↓ | FVD ↓ |
> |:-----------:|:--------:|:--------:|:---------:|:-------:|
> |    0.2    | 19.85  | 0.7632 | 0.1865  | 42.30 |
> |    0.4    | 20.12  | 0.7685 | 0.1817  | 41.32 |
> |    0.6    | 20.32  | 0.7719 | 0.1793  | 40.78 |
> |    0.8    | 20.10  | 0.7701 | 0.1806  | 41.05 |
>
>
>
> In R-Tab 1, we change the confidence threshold in the shadow detector, where a low threshold means detections with low confidence will be preserved and used in the following outpainting process.
> Usually, the less prominent shadows have low confidence, and thus can be filtered out by setting a high confidence threshold.
> Even though some of the less prominent shadows cannot be filtered out by setting a high confidence threshold, our method still performs robustly on them.
> As can be seen from R-Tab 1, our method is not sensitive to the quality of the shadow detection, and performs well even when setting the confidence threshold to be 0.2, in which situation most of the low-quality shadow detections are preserved while the outpainting quality decreases slightly.
>
> *R-Tab 2: Quantitative comparison on video in different camera motion settings from the Youtube-VOS dataset.*
> | Camera Motion | Method | PSNR ↑ | SSIM ↑ | LPIPS ↓ | FVD ↓ |
> |:---------------:|:--------:|:--------:|:--------:|:---------:|:-------:|
> |      Slow     | M3DDM  | 20.31  | 0.7351 | 0.1823  | 65.47 |
> |               | MOTIA  | 20.39  | 0.7682 | 0.1701  | 57.94 |
> |               | Ours   | 20.43  | 0.7752 | 0.1772  | 39.18 |
> |      Fast     | M3DDM  | 19.89  | 0.7197 | 0.1943  | 70.06 |
> |               | MOTIA  | 19.98  | 0.7518 | 0.1804  | 62.21 |
> |               | Ours   | 20.13  | 0.7648 | 0.1855  | 44.86 |
>
>
> The scale of the shadow distortion is hard to quantify directly since there are no ground-truth shadow annotations in our adopted datasets.
> We therefore measure the camera motions instead, in which large camera motions between two frames are more likely with largely distorted shadows.
> The inter-frame camera motion is first measured by Structure-from-Motion (SfM).
> Then, for a single video, if its average per-frame translation is less than 5 Euclidean distance and average per-frame rotation angle is less than 5 degrees, we label it as a slow-motion video, otherwise a fast-motion one.
> As can be seen from R-Tab 2, although the fast-motion videos should be with largely distorted shadows, our method performs robustly on them.
>
>
> ---------------------------------------------------
> **Q2:** The impact of the initial object-shadow detection.
>
> **R2:** As can be seen from  R-Tab 1, our method can handle the impact of initial object-shadow detection even when the confidence threshold in detection is set to be low, where lots of detected shadows with low confidence are used for outpainting.
> This is because our spatio-temporal fusion module is specifically designed to refine the initial detection results, thus enhancing the system’s robustness to errors such as imprecise masks or overlapping shadows.
>
> ---------------------------------------------------
> **Q3:** The training strategy of the modular components.
>
> **R3:** The shadow-aware instance prediction module and the instance-aware optical flow completion module are pre-trained separately.
> Then, the entire framework is trained together in an end-to-end manner.

---

> ### Author Response · Authors · 2025-08-09
>
> Dear Reviewer G7tk,
>
> We are grateful that our response has addressed your concerns and deeply appreciate your positive score for our work.
>
>
> We also thank you for your constructive feedback and will thoughtfully incorporate your comments, together with the additional analyses, into the revised manuscript.
>
> Warm regards,
>
> The Authors of Submission 19821

---

### Note · Authors · 2025-08-14

Dear Reviewers and (Senior) Area Chair,

We sincerely appreciate the time and effort you have devoted to reviewing our work and providing constructive feedback.

We are encouraged that the reviewers recognized the novelty of treating dynamic shadows as informative cues for video outpainting (G7tk, uqHS, ak5T, wJGo) and the interpretability to explicit incorporation of shadow prior relationships (G7tk, uqHS, wJGo). We also value their acknowledgement of the well-structured multi-module architecture (G7tk, uqHS) and the improvements over baseline methods in visual quality and temporal consistency (G7tk, uqHS, ak5T, wJGo).

During the discussion period, we addressed most of the reviewers’ concerns with additional experiments and clarifications, and ultimately received positive scores from all reviewers for our work.

We will incorporate all of your comments, along with the points discussed during the rebuttal, into the revised manuscript.

Once again, thank you for your dedicated efforts in reviewing our submission.

Best regards,

The authors of submission 19821

---

### Decision · Program_Chairs · 2025-09-17

**Decision:**

Accept (poster)

**Comment:**

This paper introduces a shadow-aware video outpainting framework that explicitly models object–shadow pairs to reveal the spatial extent and motion of temporarily invisible instances, coupling instance-aware optical-flow completion with a diffusion outpainting backbone and a video-aware shadow-alignment discriminator. Reviewers find the core idea novel and well-motivated, with coherent modular design and strong empirical gains (PSNR/SSIM/FVD) plus a user study; after rebuttal, all reviewers are positive (one accept, three borderline accept). The rebuttal further strengthens the case with robustness analyses (insensitivity to shadow-detector thresholds), camera-motion splits showing stability under fast motion, and an efficiency comparison indicating competitive inference without per-video test-time retraining (unlike MOTIA). Thus, the AC decides to accept the submission.